# INVARIANCE-GUIDED FEATURE EVOLUTION FOR FEW-SHOT LEARNING

## ABSTRACT

Few-shot learning (FSL) aims to characterize the inherent visual relationship between support and query samples which can be well generalized to unseen classes so that we can accurately infer the labels of query samples from very few support samples. We observe that, in a successfully learned FSL model, this visual relationship and the learned features of the query samples should remain largely invariant across different configurations of the support set. Driven by this observation, we propose to construct a feature evolution network with an ensemble of few-shot learners evolving along different configuration dimensions. We choose to study two major parameters that control the support set configuration: the number of labeled samples per class (called *shots*) and the percentage of training samples (called *partition*) in the support set. Based on this network, we characterize and track the evolution behavior of learned query features across different shots-partition configurations, which will be minimized by a set of invariance loss functions during the training stage. Our extensive experimental results demonstrate that the proposed invariance-guided feature evolution (IGFE) method significantly improves the performance and generalization capability of few-shot learning and outperforms the state-of-the-art methods by large margins, especially in cross-domain classification tasks for generalization capability test. For example, in the cross-domain test on the fine-grained CUB image classification task, our method has improved the classification accuracy by more than 5%.

## 1   INTRODUCTION

Few-shot image classification aims to learn to predict the labels of query images from very small sets of support samples (Finn et al., 2017; Yu et al., 2020). More importantly, this learned representation needs to generalize well onto unseen classes during the test stage. Since the number of support images per class, also called *shots*, is very small, often in the range of 1 to 5, the characteristics of support samples will vary significantly. Therefore, a successful few-shot learning (FSL) should be able to survive this large variation of image-specific characteristics in the support set (Yu et al., 2020; Bronskill et al., 2020). This requires that the FSL be able to capture the underlying or inherent visual relationship between the query and support images, instead of being distracted or over-fitted by the image- specific characteristics, especially those non-salient features from the background. In other words, this inherent visual relationship, once successfully learned, should remain relatively invariant across different sample configurations of the support sets. Otherwise, its classification performance with very few support samples and generalization capability onto novel classes will degrade significantly.

Based on this observation, we put forward the following observation: the feature representation generated by the FSL, if successfully learned, should remain relatively invariant to sample configurations of the support sets, for example, the number of shots (i.e, the number of samples per class) and the selection of support samples for training. Driven by this observation, we propose to construct a feature evolution network with an ensemble of few-shot learners which evolve the feature embedding along dimensions of the support set configuration, as illustrated in Figure 1. In this work, we choose to study two major parameters that control the support set configuration: the number of labeled samples per class (called *shots*) and the percentage of training samples (called *partition*) in the support set. This network aims to characterize and track the evolution and variation of learned features of the query samples across different shot-partition configuration of the support samples.

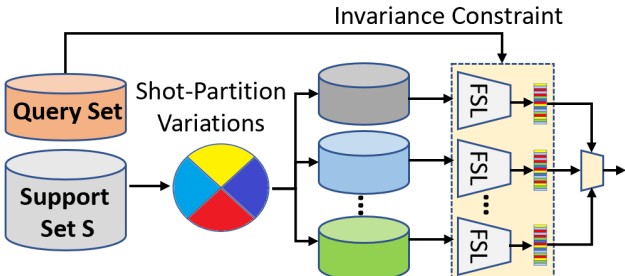

Figure 1: Illustration of the proposed idea for invariance-guided feature evolution for few-shot learning.

We then introduce an invariance metric to measure the variation of the learned query features. This invariance metric serves as a loss function to be optimized during the FSL process. Our extensive experimental results demonstrate that the proposed IGFE method significantly improves the performance of FSL and outperforms existing state-of-the-art methods by large margins on various benchmark datasets.

## 2 RELATED WORK

There are three major categories of methods that have been developed for few-shot learning: meta learning, transfer learning, and metric learning.

(1) In meta-learning based methods, episode training, which mimics the test tasks during the training stage, is explored to improve the knowledge transfer and generalization capability from meta-training to meta-testing (Finn et al., 2017; Rusu et al., 2018). In Zhang et al. (2020a); Ye et al. (2020); Chen et al. (2020b), meta-learning is combined with pre-training to further improve the performance. Tian et al. (2020); Rizve et al. (2021) decoupled the learning procedure into the base-class embedding pre-training and novel-class classifier learning and develop a multivariate logistic regression and knowledge distillation method to improve the meta-learning performance.

(2) Transfer-learning methods (Qiao et al., 2018; Gidaris & Komodakis, 2018; Chen et al., 2019) pre-train the model with a large set of base classes and then adapt the model to unseen classes. Transmatch(Yu et al., 2020) pre-trains a feature extractor on the base-class data, uses it as the feature extractor to initialize the classifier weights for unseen classes, and updates the model using semi-supervised learning methods. Gidaris & Komodakis (2018) propose an attention module to predict the classifier weights for novel classes dynamically. Instead of separating the base classes into a set of few-shot tasks, methods in Qiao et al. (2018); Chen et al. (2019) use all base classes to pre-train the few-shot model, which is then adapted to novel-class recognition.

(3) Metric-based methods (Sung et al., 2018; Zhang et al., 2020b) aim to learn an effective measurement space and identify query samples based on the feature distance between samples. MatchingNet (Vinyals et al., 2016) shows that neural networks can be successfully trained to measure the similarity between query and support samples. The prototype network method (Snell et al., 2017) introduces a prototype, which represents a class. The distance between prototypes and a query vector is used for the classification. Recent methods utilize task-specific prototypes. For example, FEAT (Ye et al., 2020) applies self-attention-based transformation on the prototypes. BSNet (Li et al., 2021) uses a dual-metric distance and achieves better performance than a single-metric distance.

Recently, graph-based methods have been explored to model and predict the correlation between samples, aiming to propagate meta-knowledge and the labels from base classes to novel classes (Rodríguez et al., 2020; Satorras & Estrach, 2018). Chen et al. (2020a) develop a graph neural network-based few-shot learning method which iteratively aggregates features from neighbors and learn the correlation between samples. DPGN (Yang et al., 2020) has proposed a two-fold complete graph network to model instance-level and distribution-level relations with label propagation and transduction. The MCGN method (Tang et al., 2021) combines GNN and conditional random field (CRF) to achieve better affinity prediction.

**Knowledge distillation.** This work is related to knowledge distillation, which learns a small student network using the supervision signals from both ground truth labels and a larger teacher network (Gou et al., 2021). Three different types of knowledge or supervision signals have been explored in the literature. (1) *Response-based knowledge distillation* uses the neural response of the last output layer of the teacher network as the supervision information (Hinton et al., 2015; Zhang et al., 2019; Meng et al., 2019). (2) *Feature-based knowledge distillation* uses the feature map of the middle layer as the supervision signal to train the student network (Romero et al., 2014; Wang et al., 2020; Chen et al., 2021). (3) *Relationship-based knowledge distillation* further explores the relationship between different layers or data samples (Yim et al., 2017; Passalis et al., 2020; Dong et al., 2021). For example, the second order statistics between different network layers are used in Yim et al. (2017) as the supervision signals.

## 3 METHOD

In this section, we present our proposed method of invariance-guided feature evolution (IGFE) for few-shot learning.

### 3.1 CLASS FEATURE VARIATIONS ACROSS DIFFERENT SUPPORT SET CONFIGURATIONS

In this section, we experimentally study the variation of query set features across different support set configurations, which motivates us to make the invariance observation discussed in Section 1. In a typical setting, a $K$-way $N$-shot FSL has $K$ classes with $N$ support samples per class in its training batch. The training samples from one class will be divided into $M$ batches. In total, we have $K \times M$ samples per class.

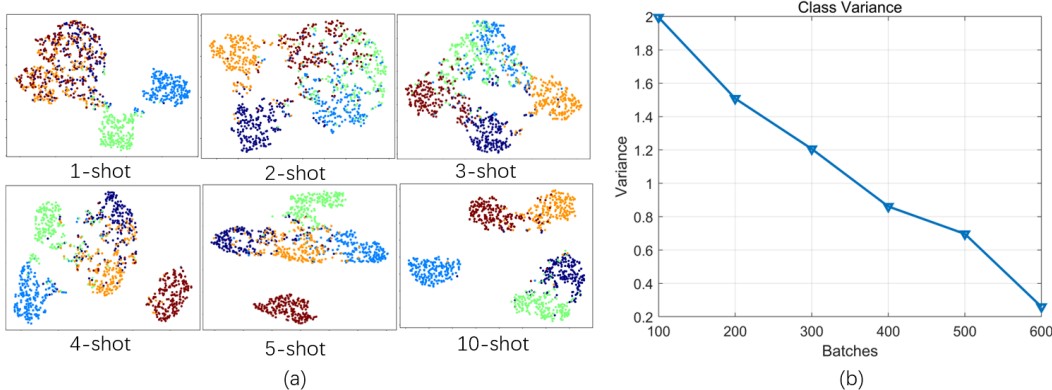

Figure 2: (a) visualization of class features across partitions for different numbers of shots; (b) average variance of class features across shots for different numbers of batches of support samples used for training.

The performance of the FSL directly depends on the configuration of the support set. In this work, we study two important parameters that plays important roles in FSL: (1) $N$, the number of shots or the number of support samples per class, and (2) $M$, the number of sample batches or the selected partition of the support set used for training. For simplicity, we refer to these types of configuration as shot configuration and partition configuration. In the following experiments, we perform few-shot learning on the mini-ImageNet dataset. Figure 2(a) shows the t-SNE plots for query sample features of 5 classes for different shots $N$. In each shot configuration, we choose different subset partitions of the support samples for training. We can see that, for FSL with larger $N$, the variation of query class features is becoming smaller. This implies that, in a better learned FSL model, the feature variation across different support set partition is becoming smaller. Figure 2(b) shows the experimental results for different numbers of batches or subset partitions of the support set used for training. For each batch or support set partition, we perform the FSL with different $N$, and compute the average variation of query class features. We can see that, as more batches of support samples are used, the FSL is better learned, the query class feature variation across different shots is becoming smaller.

These experimental results motivate us to make the following observation: *in a better learned FSL model, the variation of query sample feature across different shot and partition configurations of*

*the support will be smaller*. Driven by this observation, we develop the invariance-guided feature evolution (IGFE) method for few-shot learning, which will be explained int the following section.

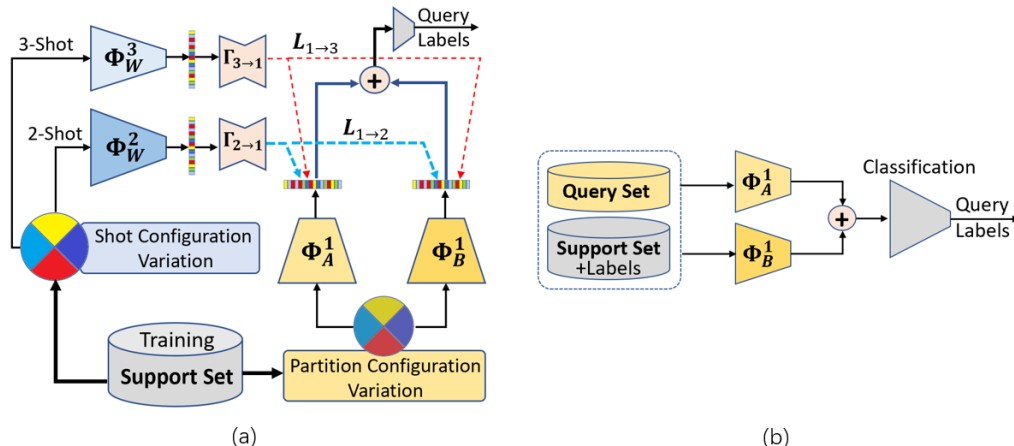

Figure 3: (a) Overview of the proposed IGFE method; (b)The IGFE testing scheme.

## 3.2 INVARIANCE-GUIDED FEATURE EVOLUTION

Figure 3(a) provides an overview of the proposed invariance-guided feature evolution (IGFE) method. we use the one-shot learning as an example to explain our proposed IGFE method. For the partition variation, we use three different subsets of the support samples for training: (1) $\mathbf{S}_W$ which is the whole support set; (2) $\mathbf{S}_A$ and $\mathbf{S}_B$ which are two randomly chosen subsets of the support set. For example, in our experiments, each subset contains 60% of the whole support samples. For the shot variation, besides the baseline one-shot learning module, we introduce two additional FSL modules with $N = 2$ (two-shot) and $N = 3$ (three-shot). In this way, we have the following four FSL modules: $\mathbf{\Phi}_W^2$ (two-shot) and $\mathbf{\Phi}_W^3$ (thee-shot) trained on the whole support set, $\mathbf{\Phi}_A^1$ (one-shot) and $\mathbf{\Phi}_B^2$ (one-shot) trained on support subsets $\mathbf{S}_A$ and $\mathbf{S}_B$, respectively. All of these four FSL modules are using the same graph neural network (GNN) design which will be explained in the following section.

In this work, we propose to explore the invariance property of the learned FSL feature across different support set configurations to optimize the generalization capability of the FSL. As illustrated in Figure 3(a), let $X$ be the input image. $\mathbf{\Phi}_A^1(X)$ and $\mathbf{\Phi}_B^1(X)$ be the features generated by the one-shot learning networks trained on support subsets $\mathbf{S}_A$ and $\mathbf{S}_B$, respectively. Let $\mathbf{\Phi}_W^2(X)$ and $\mathbf{\Phi}_W^3(X)$ be the features generated by the two-shot and three-shot networks trained on the whole support set $\mathbf{W}$, respectively. During our experiments, we observe that it is not efficient to directly measure and optimize the distance between the one-shot and two-shot features. Instead, we introduce two auto-encoder networks, $\mathbf{\Gamma}_{2\to1}$ and $\mathbf{\Gamma}_{3\to1}$ to transform the feature vectors $\mathbf{\Phi}_W^2(X)$ and $\mathbf{\Phi}_W^3(X)$, respectively, so as to match the feature vectors generated by the one-shot networks. Then, we introduce the following two feature invariance loss functions for the one-shot network $\mathbf{\Phi}_A^1$

$$\mathcal{L}_{1\to2}^A = \mathbb{E}_X\{||\mathbf{\Phi}_A^1(X) - \mathbf{\Gamma}_{2\to1}(\mathbf{\Phi}_W^2(X))||_2\}, \tag{1}$$

$$\mathcal{L}_{1\to3}^A = \mathbb{E}_X\{||\mathbf{\Phi}_A^1(X) - \mathbf{\Gamma}_{3\to1}(\mathbf{\Phi}_W^3(X))||_2\}. \tag{2}$$

Here, $\mathbb{E}_X\{\cdot\}$ represents the statistical average over the training batch. Similarly, we can define the feature variance loss $\mathcal{L}_{1\to2}^B$ and $\mathcal{L}_{1\to3}^B$ for the one-shot network $\mathbf{\Phi}_B^1$.

Note that one-shot networks $\mathbf{\Phi}_A^1(X)$ and $\mathbf{\Phi}_B^1(X)$ are trained over two different subsets of the support samples. This unique design is motivated by the following two considerations. (1) It can be used to measure and minimize the variation across the partition configuration variation, i.e., different partitions of the support set for training. (2) The fusion of the features generated by these network can result in more efficient few-shot classification with improved generalization capability. To measure and minimize the invariance between these two one-shot networks, we propose to analyze the their softmax outputs. Specifically, let $\mathbf{\Theta}_A(X) = [\theta_A^1, \theta_A^2, \cdots, \theta_A^I]$ be the output at the softmax

layer of network $\mathbf{\Phi}_A^1$. Then, the probability for sample $X$ to be in class $i$ is given by

$$p_A^i(X) = \frac{\exp(\theta_A^i)}{\sum_{i=1}^I \exp(\theta_A^i)}. \tag{3}$$

Similarly, we can define the class probability $p_B^i(X)$ for network $\mathbf{\Phi}_B^1$. We use the Kullback-Leibler (KL) Divergence to measure the invariance between the outputs of these two one-shot networks

$$\mathcal{L}_P = \sum_X \sum_{i=1}^I p_A^i(X) \log\left[\frac{p_A^i(X)}{p_B^i(X)}\right]. \tag{4}$$

In our experiments, we use the loss function $\mathcal{L}_P + \mathcal{L}_{1\to2} + \mathcal{L}_{1\to3}$ to train both one-shot networks.

It should be noted that the loss function in (4) is defined for the whole batch of samples. In other words, the training process aim to the minimize the overall variation of a set of query samples, instead of an individual one. From our experiments, we observe that, for each individual query sample, the features generated by these two one-shot networks will differ slightly. To produce the feature $\mathbf{F}^1(X)$ for each individual query sample, we propose to fuse these two features using the following average operation

$$\mathbf{F}^1(X) = \frac{\mathbf{\Phi}_A^1(X) + \mathbf{\Phi}_B^1(X)}{2}. \tag{5}$$

It should be noted that the above shot-partition configuration variations are only performed on the support set during the training stage. During the testing stage, once successfully trained, only the baseline 1-shot network is used for testing. Figure 3(b) shows the network structure in the test stage. The support samples (with labels) and the query samples (without labels) are passed to both trained networks $\mathbf{\Phi}_A^1$ and $\mathbf{\Phi}_B^1$ to produce the features for the each query sample. These two features are averaged according to (5), and then passed to the classification head to predict the query labels.

We have used the 5-way 1-shot learning as an example to explain the proposed IGFE method. It can be easily extended to other settings of few-shot learning. For example, for 5-way 5-shot, those two networks $\mathbf{\Phi}_W^2$ (2-shot) and $\mathbf{\Phi}_W^3$ (3-shot) introduced for shot variations will be replaced with 10-shot and 15-shot configurations. Note that both the shot and the partition configurations are simply different ways to use the support samples during the training stage.

### 3.3 BIPARTITE GRAPH NEURAL NETWORK FOR SUPPORT-QUERY SAMPLE ANALYSIS

In this work, we propose to use a bipartite graph neural network (GNN) to perform joint analysis of the support and query samples. In $K$-way $N$-shot learning, given $K$ classes, each with $N$ support samples $\{\mathbf{S}_{kn}\}$, we need to learn the FSL network to predict the labels for $K$ query samples $\{\mathbf{Q}_k\}$. This implies, in each training batch, we have $K \times (N + 1)$ support samples and query samples. As illustrated in Figure 4, we use a backbone network, for example, ResNet-10 or ResNet-12, to extract feature for each of these support and query samples. Denote their features by $\mathbf{S} = \{\mathbf{s}_{kn}^t\}$ and $\mathbf{Q} = \{\mathbf{q}_k^t\}$ where $t$ represents the update iteration index in the GNN. Initially, $t = 0$. These support-query sample features form the nodes for the GNN, denoted by $\{\mathbf{x}_j^t | 1 \le j \le J\}$, $J = K \times (N + 1)$, for the simplicity of notations. The edge between two graph node represents the correlation $\psi(\mathbf{x}_i^t, \mathbf{x}_j^t)$ between nodes $\mathbf{x}_i^t$ and $\mathbf{x}_j^t$. Note that our GNN has two types of nodes: support sample nodes and query sample nodes. The support samples nodes have labels while the labels of the query samples need to be predicted. If $\mathbf{x}_i^t$ and $\mathbf{x}_j^t$ are both support nodes, we have

$$\psi(\mathbf{x}_i^t, \mathbf{x}_j^t) = \begin{cases} 1 & if \ l(\mathbf{x}_i^t) = l(\mathbf{x}_j^t), \\ 0 & if \ l(\mathbf{x}_i^t) \neq l(\mathbf{x}_j^t). \end{cases} \tag{6}$$

Here $l(\cdot)$ represents the label of the corresponding support sample. Since the labels for the query nodes are unknown, the correlation for edges linked to these query need to be learned by the GNN. Initially, they can be set as random values between 0 and 1. Note that the edge connection between the query node and a support node represents the correlation between them. Therefore, we can use this correlation provide very important information for inferring the label for the query node from the label of the other support node.

Each node of the GNN combines features from these neighboring nodes with the corresponding correlation as weights and updates its own feature by learning a multi-layer perceptron (MLP) network

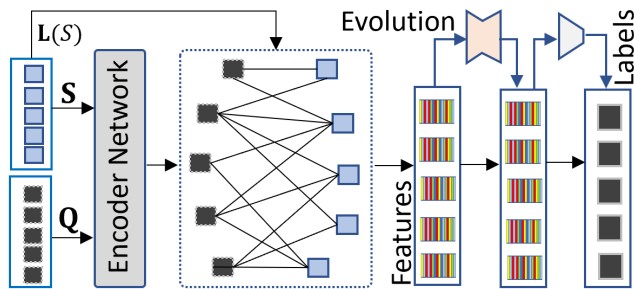

Figure 4: Overview of the graph neural network for few-shot feature embedding.

$\mathcal{F}[\cdot]$ as follows

$$\mathbf{x}_j^{t+1} = \mathcal{F}\left[\sum_{i=1}^{J} \mathbf{x}_j^t \cdot \psi(\mathbf{x}_i^t, \mathbf{x}_j^t)\right]. \tag{7}$$

At each edge, another MLP network $\mathcal{G}[\cdot, \cdot]$ is learned to predict correlation between its two nodes,

$$\psi(\mathbf{x}_i^t, \mathbf{x}_j^t) = \mathcal{G}[\mathbf{x}_i^t, \mathbf{x}_j^t], \tag{8}$$

whose ground-truth values are determined using the scheme discussed in the above. This bipartite GNN is jointly trained with the feature transform (or evolution) networks, and the final classification network which predicts the query labels.

Table 1: Classification accuracy of general few-shot classification on mini-ImageNet.

| Methods | Backbone | mini-ImageNet | |
| --- | --- | --- | --- |
| | | 5-way 1-shot | 5-way 5-shot |
| LEO (Rusu et al., 2018) | WRN-28 | $61.76 \pm 0.08\%$ | $77.59 \pm 0.12\%$ |
| PPA (Qiao et al., 2018) | WRN-28 | $59.60 \pm 0.41\%$ | $73.74 \pm 0.19\%$ |
| wDAE (Gidaris & Komodakis, 2019) | WRN-28 | $61.07 \pm 0.15\%$ | $76.75 \pm 0.11\%$ |
| CC+rot (Gidaris et al., 2019) | WRN-28 | $62.93 \pm 0.45\%$ | $79.87 \pm 0.33\%$ |
| ABNet (Yan et al., 2021) | Res-18 | $63.15 \pm 0.63\%$ | $78.85 \pm 0.56\%$ |
| ProtoNet (Chen et al., 2019) | Res-10 | $51.98 \pm 0.84\%$ | $72.64 \pm 0.64\%$ |
| MatchingNet (Chen et al., 2019) | Res-10 | $54.49 \pm 0.81\%$ | $68.82 \pm 0.65\%$ |
| RelationNet (Chen et al., 2019) | Res-10 | $52.19 \pm 0.83\%$ | $70.20 \pm 0.66\%$ |
| Cross Domain (Tseng et al., 2020) | Res-10 | $66.32 \pm 0.80\%$ | $81.98 \pm 0.55\%$ |
| FEAT (Fei et al., 2020) | Res-12 | $66.78 \pm 0.20\%$ | $82.05 \pm 0.14\%$ |
| FRN (Wertheimer et al., 2021) | Res-12 | $66.45 \pm 0.19\%$ | $82.83 \pm 0.13\%$ |
| infoPatch (Liu et al., 2021) | Res-12 | $67.67 \pm 0.45\%$ | $82.44 \pm 0.31\%$ |
| IER-Distill (Rizve et al., 2021) | Res-12 | $67.28 \pm 0.80\%$ | $84.78 \pm 0.52\%$ |
| Our IGFE Method | Res-10 | $\mathbf{71.48 \pm 0.26}\%$ | $\mathbf{86.22 \pm 0.16}\%$ |
| | Res-12 | $\mathbf{72.06 \pm 0.25}\%$ | $\mathbf{86.59 \pm 0.16}\%$ |

Table 2: Classification accuracy of fine-grained few-shot classification on CUB.

| Methods | Backbone | CUB | |
| --- | --- | --- | --- |
| | | 5-way 1-shot | 5-way 5-shot |
| BsNet(D&C) (Li et al., 2021) | Conv4 | $65.20 \pm 0.92\%$ | $84.18 \pm 0.64\%$ |
| DN4 (Li et al., 2019) | Conv8 | $64.26 \pm 1.01\%$ | $84.41 \pm 0.63\%$ |
| Baseline++ (Chen et al., 2019) | Res-18 | $67.02 \pm 0.90\%$ | $83.58 \pm 0.54\%$ |
| MatchNet (Vinyals et al., 2016) | Res-18 | $73.49 \pm 0.89\%$ | $84.45 \pm 0.5\%8$ |
| Neg-Cosine (Liu et al., 2020) | Res-18 | $72.66 \pm 0.85\%$ | $89.40 \pm 0.43\%$ |
| DSN† (Simon et al., 2020) | Res-12 | $79.96 \pm 0.21\%$ | $91.41 \pm 0.34\%$ |
| CTX† (Doersch et al., 2020) | Res-12 | $79.34 \pm 0.21\%$ | $91.42 \pm 0.11\%$ |
| FRN (Wertheimer et al., 2021) | Res-12 | $83.55 \pm 0.19\%$ | $92.92 \pm 0.10\%$ |
| Our IGFE Method | Res-10 | $\mathbf{85.47 \pm 0.22}\%$ | $\mathbf{94.33 \pm 0.10}\%$ |
| | Res-12 | $\mathbf{86.54 \pm 0.22}\%$ | $\mathbf{94.53 \pm 0.10}\%$ |

## 4    EXPERIMENTS

In this section, we conduct experiments to evaluate the performance of the proposed IGFE method and compare it with the state-of-the-art methods. We also provide ablation studies to further understand the proposed method.

### 4.1 DATASET

We evaluate our method on five widely used few-shot learning benchmarks: **mini-ImageNet** (Vinyals et al., 2016), **CUB-200-2011** (Wah et al., 2011), **Cars** (Krause et al., 2013), **Places** and **Plantae** (Van Horn et al., 2018). More details of dataset are presented in Appendix A.1.

### 4.2 IMPLEMENTATION DETAILS

We conduct experiments with two backbone networks, ResNet10 and ResNet12 (Tseng et al., 2020), which are extensively used in few-shot learning tasks. The feature transformation MLP has three hidden layers with dimensions of 480, 480 and 458, respectively. Meanwhile, normalization and activation operations are performed after each hidden layer. Here we use LeakyReLu (Xu et al., 2015) as the activation function. We use the Adam optimizer in all experiments. We first pretrain a feature extractor with 400 epochs on the base class (e.g., the mini-ImageNet) and set the classification category to 200. The GNN-based feature embedding networks are trained with 400 epochs. In all training and testing experiments, we resize the picture to 224×224.

Evaluations are conducted in 5-way 1-shot and 5-shot settings on benchmark datasets, including mini-ImageNet, CUB-200-2011, Cars, Places, and Plantae. In the $N$-way $K$-shot setting, a meta-test task has $N$ classes, each with $K$ labeled support samples. We randomly sample 10000 meta-test tasks from the test dataset and then report the mean accuracy and the 95% confidence interval. We also randomly sample 15 queries for each of the five classes in 5-way 1-shot and 5-shot settings for each meta-test task.

In order to evaluate the effectiveness of our proposed IGFE method, following existing work in the literature, we conduct FSL experiment in the following three scenarios: (1) general few-shot classification in the same object domain, (2) fine-grained few-Shot classification in the same domain, and (3) cross-Domain few-shot classification. For the cross-domain FSL, the image classes for the training and those for the testing are from different domains, for example, mini-ImageNet images for training and CUB images for testing.

**(1) General Few-Shot Image Classification.** In the first part of our few-shot learning experiments, we evaluate the FSL performance on the mini-ImageNet dataset. Table 1 shows the results of our method and existing methods in the literature. We can see that our IGFE method outperforms the current best EI-Distill method (Rizve et al., 2021) by a large margin. For example, for the 1-shot classification, our method improves the accuracy by 4.78% with the ResNet-12 backbone. For the 5-shot classification, the performance gain is 1.81%.

**(2) Fine-Grained Few-Shot Classification.** The CUB-200-2011 dataset is often used to evaluate the performance of FSL methods on fine-grained image classification. Table 2 summarizes the performance comparison results. It can be seen that our method outperforms existing state-of-the-art methods. For the 1-shot fine-grained classification task, our method improves the classification accuracy by 2.99% upon the current best FRN method (Wertheimer et al., 2021). For the 5-shot classification task, our method outperforms the FRN method by 1.61%.

Table 3: Cross-domain few-shot classification results. We train the model on the mini-ImageNet domain and evaluate the trained model in other domains.

| 5-way 1-shot | Backbone | CUB | Cars | Places | Plantae |
|---|---|---|---|---|---|
| MatchingNet+FT (Tseng et al., 2020) | Res-10 | $36.61 \pm 0.53\%$ | $29.82 \pm 0.44\%$ | $51.07 \pm 0.68\%$ | $34.48 \pm 0.50\%$ |
| RelationNet+FT (Tseng et al., 2020) | Res-10 | $44.07 \pm 0.77\%$ | $28.63 \pm 0.59\%$ | $50.68 \pm 0.87\%$ | $33.14 \pm 0.62\%$ |
| GNN+FT citeptseng2020cross | Res-10 | $47.47 \pm 0.75\%$ | $31.61 \pm 0.53\%$ | $55.77 \pm 0.79\%$ | $35.95 \pm 0.58\%$ |
| GNN+ATA (Wang & Deng, 2021)) | Res-10 | $45.00 \pm 0.50\%$ | $33.61 \pm 0.40\%$ | $53.57 \pm 0.50\%$ | $34.42 \pm 0.40\%$ |
| LPR-GNN (Sun et al. (2021) | Res-10 | $48.29 \pm 0.51\%$ | $32.78 \pm 0.39\%$ | $54.83 \pm 0.56\%$ | $37.49 \pm 0.43\%$ |
| Our IGFE Method | Res-10 | $\mathbf{49.94 \pm 0.24}\%$ | $\mathbf{34.13 \pm 0.20}\%$ | $\mathbf{60.76 \pm 0.28}\%$ | $\mathbf{39.77 \pm 0.21}\%$ |
| | Res-12 | $\mathbf{53.16 \pm 0.25}\%$ | $\mathbf{33.94 \pm 0.19}\%$ | $\mathbf{59.56 \pm 0.27}\%$ | $\mathbf{40.51 \pm 0.21}\%$ |
| 5-way 5-shot | Backbone | CUB | Cars | Places | Plantae |
| MatchingNet+FT (Tseng et al., 2020) | Res-10 | $55.23 \pm 0.83\%$ | $41.24 \pm 0.65\%$ | $64.55 \pm 0.75\%$ | $41.69 \pm 0.63\%$ |
| RelationNet+FT (Tseng et al., 2020) | Res-10 | $59.46 \pm 0.71\%$ | $39.91 \pm 0.69\%$ | $66.28 \pm 0.72\%$ | $45.08 \pm 0.59\%$ |
| GNN+FT (Tseng et al., 2020) | Res-10 | $66.98 \pm 0.68\%$ | $44.90 \pm 0.64\%$ | $73.94 \pm 0.67\%$ | $53.85 \pm 0.62\%$ |
| GNN+ATA (Wang & Deng, 2021) | Res-10 | $66.22 \pm 0.50\%$ | $49.14 \pm 0.40\%$ | $75.48 \pm 0.40\%$ | $52.69 \pm 0.40\%$ |
| LPR-GNN (Sun et al., 2021) | Res-10 | $64.44 \pm 0.48\%$ | $46.20 \pm 0.46\%$ | $74.45 \pm 0.47\%$ | $54.46 \pm 0.46\%$ |
| Our IGFE Method | Res-10 | $\mathbf{69.12 \pm 0.21}\%$ | $\mathbf{49.43 \pm 0.21}\%$ | $\mathbf{78.01 \pm 0.20}\%$ | $\mathbf{59.99 \pm 0.21}\%$ |
| | Res-12 | $\mathbf{71.11 \pm 0.21}\%$ | $\mathbf{49.72 \pm 0.21}\%$ | $\mathbf{77.58 \pm 0.20}\%$ | $\mathbf{59.88 \pm 0.20}\%$ |

**(3) Cross-domain few-shot classification.** Cross-domain experiments aim to evaluate the generalization capability of few-shot learning (Chen et al., 2019). Following existing work in the literature, we train the model on base classes of the mini-ImageNet domain and evaluate the cross-domain performance of the model on the CUB, Cars, Places, and Plantae domains respectively. Table 3(top) shows the performance comparison for 5-way 1-shot image classification with existing methods. In the second column we also include the intra-domain few-shot classification accuracy on the mini-ImageNet. We can see that our IGFE method is able to significantly improve the cross-domain classification accuracy and demonstrates good generalization capability. For example, on the Places dataset, the accuracy is improved by 4.99% upon the GNN+FT (Tseng et al., 2020) method, which is quite significant. For other datasets, the performance improvement is between 1.4% and 2.2%, which is quite consistent. We also include results of our IGFE method with the ResNet-12 backbone. Table 3(bottom) shows the performance comparison for the 5-way 5-shot image classification task. Our method outperforms the current state-of-the-art methods by large margins.

## 4.3 ABLATION STUDIES

In this section, we conduct ablation studies to further understand the proposed IGFE method and evaluate the performance contributions of major algorithm components. In our IGFE method, we have introduced two major support set configurations: shot and partition configurations. The shot configuration determines the number of samples assigned to each training classes in the support set. Partition configuration determines the subset of support samples assigned to each FSL module. It should be noted that this shot and partition configurations are only performed for the training samples. In the test stage, once the base FSL module is successfully learned, these two configurations of the support set are removed. For example, in the 5-way 1-shot learning, although 2-shot and 3-shot configurations of the support set are used during the training, in the test stage, we only use the 5-way 1-shot network to predict the labels of query samples, exactly following the same protocol as in existing methods in the literature. Table 4 summarizes the performance contributions of these two algorithm components for 5-way 1-shot (top) and 5-way 5-shot (bottom) classification tasks on five benchmark datasets. the first row shows the classification accuracy of the baseline method (Tseng et al., 2020) on which we have implemented our algorithm. The second row shows the performance with the shot variation with 1-shot, 2-shot, and 3-shot configurations. The third row shows the performance with both shot and partition variations where two different partition or subsets of the support set are used to train two 1-shot networks, both guided by the shot variations. We can see that both algorithm components contribute significantly to the overall performance. However, on different datasets, their portions of contribution are vary slightly. Experiments for the 5-way 5-shot classification task are shown in Table 4(bottom). We can see that, for 5-shot classification, the partition variation has more significant contribution than the shot configuration. This is the because the number of shots in this 5-shot classification task is already large (5). So, the importance of introducing extra shot variation degrades.

In shot variation, we have included 1-shot, 2-shot, and 3-shot. It is observed that, if more shot variations are used, the performance will be further improved. For example, in Table 5, we compare the 5-way 1-shot classification accuracy with two shot variations (1-shot + 2-shot) and three shot variations (1-shot + 2-shot + 3-shot). We can see that the three shot variations scheme obtains better classification accuracy than the two shot variations scheme.

Table 4: The ablation study on mini-ImageNet, CUB, Cars, Places, and Plantae datasets. We train the model on the mini-ImageNet domain and evaluate the trained model on other domains.

| 5-way 1-shot | mini-ImageNet | CUB | Cars | Places | Plantae |
|---|---|---|---|---|---|
| Baseline | $66.32 \pm 0.80\%$ | $47.47 \pm 0.75\%$ | $31.61 \pm 0.53\%$ | $55.77 \pm 0.79\%$ | $35.95 \pm 0.58\%$ |
| + Shot Variation | $69.63 \pm 0.25\%$ | $47.86 \pm 0.21\%$ | $33.20 \pm 0.16\%$ | $58.44 \pm 0.25\%$ | $39.53 \pm 0.17\%$ |
| + Shot and Partition Variation | $\mathbf{71.48 \pm 0.26}\%$ | $\mathbf{49.94 \pm 0.24}\%$ | $\mathbf{34.13 \pm 0.20}\%$ | $\mathbf{60.76 \pm 0.28}\%$ | $\mathbf{39.77 \pm 0.21}\%$ |

| 5-way 5-shot | mini-ImageNet | CUB | Cars | Places | Plantae |
|---|---|---|---|---|---|
| Baseline | $81.98 \pm 0.55\%$ | $66.98 \pm 0.68\%$ | $44.90 \pm 0.64\%$ | $73.94 \pm 0.67\%$ | $53.85 \pm 0.62\%$ |
| + Shot Variation | $83.84 \pm 0.17\%$ | $64.03 \pm 0.21\%$ | $46.24 \pm 0.21\%$ | $74.12 \pm 0.21\%$ | $54.46 \pm 0.20\%$ |
| + Shot and Partition Variation | $\mathbf{86.22 \pm 0.16}\%$ | $\mathbf{69.12 \pm 0.21}\%$ | $\mathbf{49.43 \pm 0.21}\%$ | $\mathbf{78.01 \pm 0.20}\%$ | $\mathbf{59.99 \pm 0.21}\%$ |

Table 5: The IGFE method with different shot configurations.

| Shot Configurations | mini-ImageNet | CUB | Cars | Places | Plantae |
|---|---|---|---|---|---|
| (1-shot + 2-shot) Variation | $68.92 \pm 0.25\%$ | $47.16 \pm 0.22\%$ | $32.70 \pm 0.16\%$ | $57.83 \pm 0.25\%$ | $38.91 \pm 0.18\%$ |
| (1-shot + 2-shot + 3-shot) Variation | $69.63 \pm 0.25\%$ | $47.86 \pm 0.21\%$ | $33.20 \pm 0.16\%$ | $58.44 \pm 0.25\%$ | $39.53 \pm 0.17\%$ |

## 5 UNIQUE CONTRIBUTIONS AND FURTHER DISCUSSIONS

Compared to the existing work in the literature, the unique contributions of this work can be summarized as follows: (1) This work is motivated by an important observation that, in a successfully learned FSL model, the generated features for query samples should remain largely invariant across different support sample configurations. We then construct a network structure to characterize their feature evolution and explore their variations. (2) We introduce two types of configuration variations, namely, shot and partition variations, to construct different configurations of the support samples during the training stage. Using the GNN, we explore the relationship between the query samples and these different configurations of the support samples. We introduce an invariance loss function to train this network to minimize the feature variation across different support sample configurations. (3) Our experimental results have demonstrated that the proposed IGFE method is able to significantly improve the FSL performance, outperforming existing state-of-the-art methods by large margins. The ablation studies have also demonstrated that the proposed shot-partition configuration variation of the support set and the invariance loss on the query samples are able to significantly improve the few-shot classification accuracy upon the baseline GNN+FT method (Tseng et al., 2020), on which our algorithm is implemented.

Successful generalization of the network model learned from the training set to the test set is an important problem in machine learning (Zhou, 2014). It has been recognized that optimizing the marginal distribution by maximizing the marginal mean and minimizing the marginal variance simultaneously can lead to better generalization performance for traditional machine learning method, such as support vector machine (SVM) classification (Zhou, 2014; Zhang & Zhou, 2020). In this work, we explore the invariance constraint to minimize the feature variance of query classes so as to improve the generalization capability of the few-shot classification. From the experimental results, We can see that our proposed performs much better on cross-domain tasks than intra-domain tasks.

The proposed invariance loss across multiple networks is related to the consistency loss (Ouali et al., 2020; Abuduweili et al., 2021) and the student-teacher or dual student network (Ke et al., 2019) that have been extensively used in semi-supervised learning research. In this work, we explore this invariance and mutual consistency in the few-shot learning framework where the feature invariance or consistency is enforced between the query samples with different configurations of support samples over a graph neural network.

## 6 CONCLUSION

In this work, we have developed a new few-shot learning method, called invariance-guided feature evolution (IGFE), aiming to characterize the inherent visual relationship between support and query samples which can be well generalized to unseen classes. From our experimental studies, we have observed that, in a successfully learned FSL model, this visual relationship and the learned features of the query samples should remain largely invariant across different configurations of the support set. Driven by this observation, we have constructed a feature evolution network with an ensemble of few-shot learners evolving along different configuration dimensions. We have introduced the shot and partition configurations that control different assignments of the support samples during training. Using a bipartite graph neural network, we analyze the feature relationship between query samples and different configurations of support samples. We characterize and track the evolution behavior of learned query features across different shots-partition configurations, which is minimized by a set of invariance loss functions during the training stage. Our extensive experimental results demonstrate that the proposed invariance-guided feature evolution (IGFE) method significantly improves the performance and generalization capability of few-shot learning and outperforms the state-of-the-art methods by a large margin.

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

# A  APPENDIX

In this appendix, we provide more details of experimental settings and additional results for further understanding of the proposed method.

## A.1  DATASETS

We evaluate our method on five widely used few-shot learning benchmarks. **(1) mini-ImageNet** (Vinyals et al., 2016) is a subset of ImageNet, which includes 100 classes and each class consists of 600 images. We follow the standard protocol that utilizes 64 classes for training, 16 classes for validation, and 20 classes for testing. **(2) CUB-200-2011** (Wah et al., 2011) is a fine-grained classification dataset, which includes 200 classes and contains 11,788 images. Following Vinyals et al. (2016), we randomly split the dataset into 100 classes for training, 50 classes for validation, and 50 classes for test, respectively. **(3) Cars** (Krause et al., 2013) created by Stanford University contains 16,185 images of 196 types of cars. We randomly selected 196 categories include 98 training, 49 validation and 49 testing for the experiment. **(4) Places** (Krause et al., 2013) is a dataset of scene images, containing 73,000 training images from 365 scene categories, which are divided into 183 categories for training, 91 for validation and 91 for testing. **(5) Plantae** (Van Horn et al., 2018) is a sub-set of the iNat2017 dataset, which contains 200 types of plants and a total of 47,242 images. We split them into 100 classes for training, 50 for validation, and 50 for testing. The number of training, validation, testing categories for each dataset are summarized in Table 6.

Table 6: Summarization of the datasets (domains).

| Datasets | mini-ImageNet | CUB | Cars | Places | Plantae |
|---|---|---|---|---|---|
| Source | Vinyals et al. (2016) | Wah et al. (2011) | Krause et al. (2013) | Krause et al. (2013) | Van Horn et al. (2018) |
| Training categories | 64 | 100 | 98 | 183 | 100 |
| Validation categories | 16 | 50 | 49 | 91 | 50 |
| Testing categories | 20 | 50 | 49 | 91 | 50 |
| Split setting | Tseng et al. (2020) | Vinyals et al. (2016) | randomly split | randomly split | randomly split |

## A.2  EXTENSION TO $N$-SHOT IMAGE CLASSIFICATION

In the main paper, we used the 5-way 1-shot as an example to present the method of invariance-guided feature evolution (IGFE). This is method can be naturally extended to generic $K$-way $N$-shot image classification scenarios. Note that the shot is just a way to partition the support samples during the training stage. For example, for a 5-way 5-shot image classification, we construct a 10-shot network and a 15-shot network to guide the training of the 5-shot network. Once successfully trained, in the test stage, on the 5-shot network is used for predicting the query labels.

## A.3  VARIANCE OF QUERY SAMPLE FEATURES

As discussed in the paper, the proposed IGFE method aims to minimize the feature variance across different shot and partition configurations of the support samples. Figure 5 shows 4 examples from the mini-ImageNet few-shot image classification. The second row shows the t-SNE visualization of the query sample features from 5 query classes. Each class has 100 samples. The first row shows the query features obtained by the baseline GNN+FT (Tseng et al., 2020) method upon which our IGFE method was implemented. We can see that the IGFE is able to generate much more aggregated and compact feature clusters for the query classes than the baseline method. This leads to significant improvement of accuracy and generalization capability in few-shot image classification.

Figure 6 plots the feature variance of each query class. Each sub-figure shows one experiment of 5-way 1-shot image classification on the mini-ImageNet dataset with 100 query samples. In each

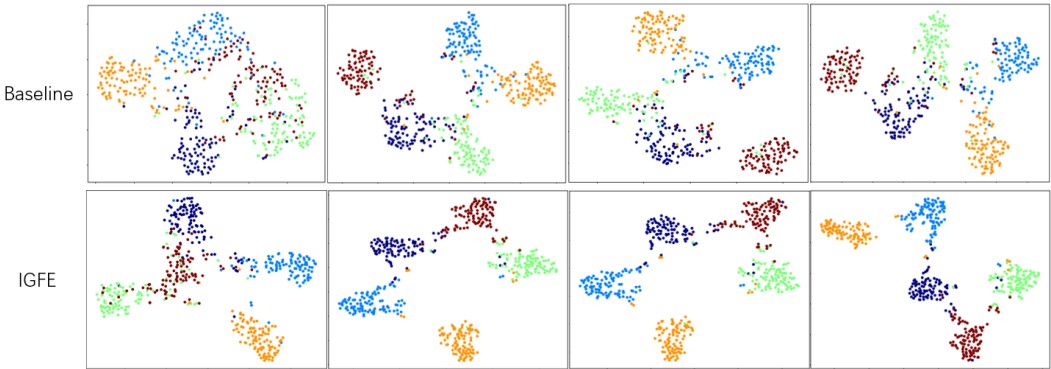

Figure 5: Visual comparison of 5-shot model on miniImageNet dataset, select 100 query samples for each category.

sub-figure, the dashed line represents the variance for each of these 5 query classes generated by the baseline method. The solid line shows the variance of our IGFE method. We can see that our method is able to significantly reduce the feature variance of the query classes.

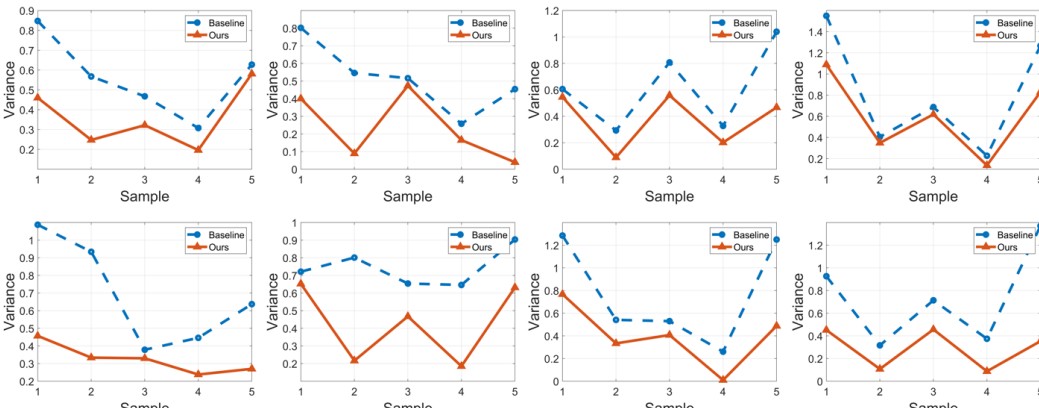

Figure 6: Variance comparison of 5-shot model on miniImageNet dataset.

