# OpenReview forum: "Invariance-Guided Feature Evolution  for  Few-Shot Learning"
_ICLR.cc/2022/Conference — ICLR 2022 Submitted_

### Official Review · Reviewer_dXjh · 2021-10-29

**Correctness:** 3
**Technical Novelty And Significance:** 2
**Empirical Novelty And Significance:** 2
**Recommendation:** 5
**Confidence:** 4

**Main Review:**

Strenghts:
This method simultaneously utilizes four FSL modules with different shot-partition configurations to train a GNN model that perform joint analysis of the support and query samples.
Besides, they propose a feature invariance loss function which reduce the variance among different configurations efficiently. The experiments show the efficiency of the proposed IGFE.

Weaknesses:
1. Some definitions are not clearly. Firstly, the two configurations considered in this paper is the shots and the partition, where the partition is the percentage of training samples in the support set, but I do not quite understand the M, the number of sample batches or the selected partition of the support set used for training, which is mentioned in Section 3. Is the M the number of samples in each class that can be used as support set in all episodes (e.g., if M=300 and there are 600 samples in each class, the support set of this class in each episode is selected in these 300 samples)? And whether the query set is selected in these M samples or in all samples of each class? If so, the expression “In total, we have K*M samples per class.” is not right. And the interpretation of M should be clearer.
2. The efficiency of GNN module should be introduced in experiments. This paper does not show the experimental results with other modules. Therefore, it’s not convictive to show the efficiency of IGFE. What’s the performance if only utilizing the linear function?
3. The efficiency of the feature invariance loss function is not introduced in experiments. Why it is not efficient to directly measure and optimize the distance between the one-shot and two-shot features?
4. The GNN module is a transductive method. What’s the performance when employing the inductive setting (Constructing each graph with only one query sample)?
5. The number of base classes is 64 in mini-ImageNet. Why to set the classification category to 200?
6. There are some typos. For example, the last sentence in Section 3.1, “which will be explained int the following section”, the “int” should be “in”. And “Φ_A^1(one-shot) and Φ_B^2(one-shot) trained on support subsets S_A and S_B”, the “Φ_B^2” should be “Φ_B^1”.


**Summary Of The Paper:**

This paper experimentally studied the variation of query set features across different support set configurations and observed that in a successfully learned FSL model, the visual relationship between support and query samples and the learned features of the query samples should remain largely invariant across different configurations of the support set. Motivated by this, this paper proposed an invariance-guided feature evolution (IGFE) method to maintain the invariance across (a) different number of labeled samples per class (called shots) and (b) different percentage of training samples (called partition) in the support set to promote the performance of FSL.

**Summary Of The Review:**

The contributions are increamental, and the writing could be improved.

---

### Official Review · Reviewer_wbRK · 2021-11-01

**Correctness:** 3
**Technical Novelty And Significance:** 2
**Empirical Novelty And Significance:** 2
**Recommendation:** 3
**Confidence:** 4

**Main Review:**

Pros:
+ I think this paper puts forward an interesting question: how does the FSL configuration affect the feature distribution? Studying this question could potentially guide the how to design an effective few-shot learner.

+ This paper provides code, which makes the paper more reproducible.

Cons:
- I am extremely confused by the motivation/idea proposed in this paper. This paper is not quite easy to follow. There are several modules proposed in this paper but the motivations behind the proposed modules are not clear. Why does this paper split the training set into 4 pieces? Why are 1 & 2 & 3 shots selected?

- It seems that the main novelty of this paper is to use GNN to refine features and make them invariant to hyper-parameters such as number of shots. However, this has been studied extensively in prior arts. What's the difference between the propose module and the previous GNN based few-shot learners?

- What does the evolution mean in the proposed GNN module? Is it the forward of the GNN module?

- The evaluation to existing methods is not fair. Most of the methods of comparison use images of 84x84 (or 80x80)  as inputs while this paper resizes all images to 224x224. So I am not able to interpret the Table 1&2&3 and to assess whether the proposed method is effective. Can authors provide the results under the same training protocols with other methods?

**Summary Of The Paper:**

This paper studies feature variance in a FSL model. It observes that support set configuration can greatly affect the feature distribution. To that end, this paper proposes modules based on GNN to learn features invariant to hyper-parameters. Specifically, this method subsamples training set into 4 subsets and each subset is encoded by the same backbone network + GNN. Then, feature invariance is enforced across subsets. By doing so, this method learns features that are invariant to both number of shots and the percentage of training samples, which are beneficial to few-shot learning evaluation.

**Summary Of The Review:**

My assessment is based on the three aspects: motivation, novelty, and results. The motivations behind proposed modules in this paper do not make sense to me and the novelty of this paper is limited. Most importantly, the comparison to previous methods is not fair. So I recommend rejection of this paper.

---

### Official Review · Reviewer_ZKYs · 2021-11-02

**Correctness:** 2
**Technical Novelty And Significance:** 2
**Empirical Novelty And Significance:** 2
**Recommendation:** 5
**Confidence:** 4

**Main Review:**

Strength:

- The problem statement of the paper to build a training strategy with an adapted architecture to increase the generalization looks interesting.
- The paper is well written.


Weaknesses:

- Unfortunately, I had some difficulties with the illustration of the problem in the beginning section of section 3 (Method). While understanding the reason for having an invariant model for meta-learning, I am not sure if the paper accurately describes the problem. The obtained tsne results applied on a pertained depends on several *missing factors* such as backbone, loss function, and a number of epochs on the tested dataset miniImagNet. For example, tsne visualizing on an overfitted deep backbone such as the ResNet (trained with many iterations on the dataset) using tSNE would result in discriminative feature embedding for 1-shot if we are visualizing the training samples.

- Applying graph neural networks at top of ResNet and testing on the small datasets might increase the generalization due to the over-parameterization. Therefore, I recommend the author to discuss the number of parameters in their proposed method and ablate the effect of the applied extra-module. For example, what happens if we train the model with a cross-entropy loss with a hybrid ResNet12-GNN model.


- I totally respect the effort of the author by evaluations with miniImageNet and CUB and other small datasets.  However, large few-shot benchmarks are required to measure the effectiveness of the method

- MiniImageNet dataset contains images of size 84x84 which is used in few-shot literature. Could you please justify the reason for using 224x224?


**Summary Of The Paper:**

This work first illustrates a problem in episodic pretraining. Particularly, the paper claims that as we increase the number of shots in the episodic pertaining, the feature space becomes more discriminative because low-shot-based training is not invariant to the extracted features of the support set. Then, the paper proposes to use an extra trainable graph neural network module to extract “invariant features”. Finally, the paper evaluates the proposed method using miniImageNet, CUB datasets using ResNet10 and ResNet12.

**Summary Of The Review:**

I think the illustration of the problem discussed in the paper is not clear, and part of the evaluation is not clear enough (please see above for the full review).

---

### Official Review · Reviewer_yhon · 2021-11-03

**Correctness:** 4
**Technical Novelty And Significance:** 3
**Empirical Novelty And Significance:** 2
**Recommendation:** 6
**Confidence:** 5

**Main Review:**

Pros:
1.	The experiments on features with different configurations are insightful.

Cons:
1.	The relationship of different shot number has been studied in the former work [1]. The authors can have some discussion about the difference.
2.	The main problem comes with the implementation of GNN. Briefly speaking, the authors implement GNN in such a way: For a N-way K-shot episode with M query samples for each class, the GNN is used M iterations, with each iteration one query sample from each class and all support samples are used as input. This leads to two problems:
a)	This procedure seems to be a transductive process since GNN can gather information from multiple query samples instead of just one sample which is the definition of inductive setting. This means it is unfair to some extent to compare the model with former inductive methods. Besides, the proposed method is not comparable with the state-of-the-art transductive methods.
b)	I am not sure if such a design can help model learn the prior knowledge that query samples in each iteration have different labels, which can be a kind of information leakage.
The authors can explain about the above questions. In my opinion, it would be better if the authors can provide results without GNN as an ablation study to see if the proposed method is truly effective.

[1]  Regularising Knowledge Transfer by Meta Functional Learning. IJCAI 2021




**Summary Of The Paper:**

This paper mainly targets few-shot learning. The authors propose to enhance the invariance of features among different configurations. They merge the regularization with GNN architecture to achieve good results on several datasets.

**Summary Of The Review:**

This paper provides insightful method and strong results. However I have questions about the implementation for now, which makes the proposed method not reliable enough.

---

> ### Author Response · Authors · 2021-11-19
> **Really appreciate your valuable review comments!**
>
> We sincerely thank you for your positive review of our paper and your valuable comments for us to revise and improve the paper!
> We will carefully follow your comments to improve our final paper. Here are our detailed responses to your comments:
>
> $\textbf{1. For your positive comments.}$ Thank you for your encouraging comments about our paper: "The experiments on features with different configurations are insightful." "This paper provides insightful method and strong results."
>
> $\textbf{2. For the comment on related work.}$ Thanks for this valuable comment! In the final paper, we will include detailed discussion about the difference between our paper the Ref. [1]. In this paper, we are exploring the invariance between different shot and partition configurations of the training process to improve the learning performance and generalization capability of the few-shot learning. We will include the above discussion in the final paper.
>
> $\textbf{3. For the comment on transductive and inductive settings.}$ Thanks for your insightful comment! In this paper, we use the GNN+FT paper (ICLR 2020) as our baseline to implement our new idea of invariance guided feature evolution and demonstrated significant performance improvement under the transductive few-shot learning setting. Following your comments, we have also tried our method under the inductive setting and also achieved significant performance improvement (about 1-3%). We will include this additional results in the final paper.
>
> Thanks again for your positive review of our paper and for your very helpful comments!

---

### Decision · Program_Chairs · 2022-01-20

**Decision:**

Reject

**Comment:**

Three out of four of the reviewers are leaning (weakly or strongly) towards rejecting this paper. Unfortunately, the authors only responded to the concerns of the most positive reviewer, making it difficult to disregard the concerns from the three more negative reviewers.

I also took a look at the paper myself, and share a number of the reviewers' concerns. First, the proposed method appears to be performing transductive inference for its predictions, while many baselines it compares with rely on inductive inference. Transductive inference is generally known to outperform inductive inference, therefore some of the improvements in accuracy can potentially be accounted to that. The authors did mention in their one author response that they generated results in the inductive setting and still saw an improvement, however the submission was not updated with details around that new experiment, making it hard to rely on it. Second, the paper is using a 224x224 resolution for images, while the original mini-ImageNet benchmark (and the majority of baselines evaluated on it) assume a 84x84 resolution. Here too, using the former resolution is known to outperform the latter. Third, I too found the paper to lack clarity at a number of places in the writing.

I also notice that the final predictions is made following the averaging of features from two models (A and B, as in Eq. 5). This is a form of model ensembling, which generally is a principle know to help improve generalization. It seems appropriate to wonder whether the baselines are worse partly due to not relying on any ensembling at all.

Finally, I've found a recent method from ICJAI 2021 (Cross-Domain Few-Shot Classification via Adversarial Task Augmentation) which appears to beat the proposed method in the cross-domain setting for the majority of domains.

Given the above, and the lack of rebuttals to the reviewers with the most concerns, I'm afraid I must recommend this paper be rejected at this time.